# Development and Validation of a New Home Cooking Frequency Questionnaire: A Pilot Study

**DOI:** 10.3390/nu14061136

**Published:** 2022-03-08

**Authors:** Leticia Goni, Mario Gil, Víctor de la O, Miguel Ángel Martínez-González, David M. Eisenberg, María Pueyo-Garrigues, Maria Vasilj, Lucía Gayoso, Usune Etxeberria, Miguel Ruiz-Canela

**Affiliations:** 1Department of Preventive Medicine and Public Health, IdiSNA, University of Navarra, 31008 Pamplona, Spain; lgoni@unav.es (L.G.); vdelao@unav.es (V.d.l.O.); mamartinez@unav.es (M.Á.M.-G.); mvasilj@alumni.unav.es (M.V.); 2CIBER Fisiopatología de la Obesidad y Nutrición (CIBERobn), Instituto de Salud Carlos III, 28029 Madrid, Spain; 3Department of Medicina Preventiva, Hospital Universitario Fundación Alcorcón (HUFA), 28922 Madrid, Spain; mario.gil.conesa@gmail.com; 4Department of Nutrition, Harvard TH Chan School of Public Health, Boston, MA 02115, USA; deisenbe@hsp.harvard.edu; 5Department of Community, Maternity and Pediatric Nursing, School of Nursing, University of Navarra, 31008 Pamplona, Spain; mpueyo.3@unav.es; 6BCC Innovation, Technology Center in Gastronomy, Basque Culinary Center, 20009 Donostia-San Sebastián, Spain; lgayoso@bculinary.com (L.G.); uetxeberria@bculinary.com (U.E.); 7Basque Culinary Center, Faculty of Gastronomy Sciences, Mondragon Unibertsitatea, 20009 Donostia-San Sebastián, Spain

**Keywords:** culinary medicine, culinary nutrition, home cooking, culinary habits, validity

## Abstract

Home cooking and the type of cooking techniques can have an effect on our health. However, as far as we know, there is no questionnaire that measures in depth the frequency and type of cooking techniques used at home. Our aim was to design a new Home Cooking Frequency Questionnaire (HCFQ) and to preliminarily assess its psychometric properties. For this purpose we used a five-phase approach, as follows: Phase 1: item generation based on expert opinion, relevant literature and previous surveys; Phase 2: content validity assessed by experts for relevance and clarity (epidemiologists, dietitians, chefs); Phase 3: face validity and inter-item reliability; Phase 4: criterion validity using a 7-day food and culinary record; and Phase 5: test stability and inter-item reliability. The content validity index for scale and item level values provided evidence of the content validity for relevance and clarity. Criterion validity analysis showed intraclass correlation coefficients ranged from 0.31–0.69. Test–retest reliability coefficients ranged from 0.49–0.92, with ƙ values > 0.44. Overall Cronbach’s alpha was 0.90. In conclusion, the HCFQ is a promising tool with sound content and face validity, substantial criterion validity, and adequate reliability. This 174-item HCFQ is the first questionnaire to assess how often people cook and which cooking methods they use at home.

## 1. Introduction

Home cooking has been associated with the quality of the diet and the promotion and maintenance of good health. Observational studies suggest a positive association between home cooking and healthy dietary patterns (Healthy Eating Index, Mediterranean diet (MedDiet)), a higher consumption of healthy food groups (fruits, vegetables), and a lower consumption of unhealthy food groups (sugar-sweetened beverages, fast food) [1]. Intervention studies aimed at improving cooking skills have shown that participants increase the frequency and variety of the consumption of vegetables and fruits and they experience sizable health improvement in terms of weight control and normalization of lipid and glucose metabolism biomarkers [2,3,4,5]. Interventions specifically addressing home cooking can also be an excellent strategy to promote the prevention of chronic diseases such as type 2 diabetes (T2D) [6,7].

Furthermore, different studies have shown that certain cooking techniques can affect health status [8,9]. The formation of certain hazardous chemicals during cooking processes, including heterocyclic aromatic amines (HAAs), polycyclic aromatic hydrocarbons (PAHs), nitrosamines, and advanced glycation end products (AGEs), can promote inflammation and oxidative stress, two of the underlying mechanisms of non-communicable chronic diseases [10,11,12,13,14]. These compounds are increased by cooking at high temperature, a longer time of heat application, lower moisture of food, the presence of trace metals, and higher pH of the cooking system. Thus, cooking techniques such as grilling, broiling, roasting, searing, and frying generate significantly more HAAs, PAHs, nitrosamines, and AGEs compared to boiling, poaching, stewing, and steaming [10,11,15,16,17]. Therefore, diet quality depends on both food choices and cooking techniques.

A number of questionnaires have been developed and validated to measure food and cooking skills [18,19,20,21,22,23,24,25,26]. However, these questionnaires were not specifically designed to measure the frequency of exposure to different cooking techniques. In fact, only five out of nine questionnaires asked about cooking frequency, with a low number of questions about this topic, ranging from one to eight, and referring to different time periods [18,19,23,25,26]. Accordingly, it may be relevant to develop a questionnaire that provides detailed information on how often people cook at home and what cooking techniques are used to prepare the different food groups (vegetables, meat, fish, eggs, pulses, and cereals).

In this pilot study we aimed to develop and preliminarily validate a new tool to evaluate cooking habits, including the techniques used when cooking at home.

## 2. Materials and Methods

The development and preliminary validation of the new Home Cooking Frequency Questionnaire (HCFQ) was carried out over five phases: questionnaire development, content validity, face validity and internal consistency, criterion validity, and test stability and internal consistency (Figure 1) [27].

This study was conducted according to the guidelines laid down in the Declaration of Helsinki, and all procedures involving research study participants were approved by the Research Ethics Committee of the University of Navarra. Written informed consent was obtained from participants in Phase 4, where identifiable information was required. Participants in Phases 1, 2, 3, and 5 were informed about the study, and they agreed to answer the questionnaires; written informed consent was not legally required in this phases of the study, since we only obtained anonymized data.

### 2.1. Phase 1: Questionnaire Development

The criteria followed to develop the first version of the questionnaire were: to measure the frequency of cooking and consumption of home-cooked food, to identify the main cooking techniques used for each food group, and to determine the main cooking habits in relation to aspects such as food shopping, menu planning, the use of different cooking utensils, and the spices and types of fats used.

A first version of the questionnaire was based on expert opinion (one dietitian-nutritionist and one epidemiologist) and a literature search focused on other previous culinary behavior questionnaires and on the relationships between culinary habits and their impact on health (Appendix A).

To minimize measurement error, short questions in closed format were designed to be concise and clearly understood. Definitions and examples were also provided to better clarify the aim of the questionnaire.

### 2.2. Phase 2: Content Validity

This phase consisted of ensuring that the item bank measured the expected construct of interest [28]. An independent panel of 6 experts participated in this initial phase, including one epidemiologist, 3 dietitian-nutritionists, and 2 chefs (Appendix A) [29]. Each expert was sent a copy of the first version of the HCFQ by email and then asked for relevance and clarity criteria, using two 4-point Likert scales [30]. A content validity index (CVI) was estimated for each domain: S-CVI/Ave—scale-level with averaging calculation method; acceptable limit >0.80 [31]; S-CVI/UA—scale-level with universal agreement method; acceptable limit >0.80 [32]. In addition, a CVI was calculated for each individual item (I-CVI; acceptable limit > 0.83) and the modified kappa concordance index (κ* 0.75–1.00 excellent, κ* 0.60–0.74 good, κ* 0.40–0.59 fair, and κ* < 0.40 poor; acceptable limit > 0.60) [29,33,34]. Items that did not meet the minimum standards were reconsidered for modification or elimination. Experts could also provide additional comments and suggestions for each item of the HCFQ. From this step, the second version of the HCFQ was obtained.

### 2.3. Phase 3: Face Validity and Internal Consistency

A quantitative and qualitative pilot study was carried out for item analysis to assess the suitability of items for inclusion in the questionnaire through face validity and internal consistency [28,35]. A convenience sample of 17 adults was recruited, balanced by sex, with a wide range of ages and with different levels of cooking experience at home. The number of subjects was based on previous similar studies [19,23,36].

Data were collected in May 2020. Participants completed the second version of the HCFQ and an ad hoc short questionnaire with eight dichotomous questions (Yes/No) and a blank space, to provide additional feedback on the new questionnaire in terms of ease of completion and comprehension, among other aspects. For face validity, frequencies were calculated, and open narrative responses about opinions and/or improvement of the instrument were analyzed for common ‘themes’ [37]. Internal consistency was determined using the Cronbach alpha coefficient for each dimension (acceptable limit > 0.70) [38]. As a result from this step, the third version of the HCFQ was obtained.

### 2.4. Phase 4: Criterion Validity

Criterion validity is defined as the degree of agreement between the new questionnaire and another measure of the construct that serves as a gold standard [39]. A 7-day food and culinary record was used as the gold standard. In this phase, participants were part of a randomized intervention pilot study, which aimed to analyze the potential health benefit effects of a home cooking intervention on patients with T2D. A total of 30 subjects were invited to participate. Data was collected in October 2020. Both the HCFQ and the gold standard were administered before starting the culinary intervention.

Volunteers wrote down the cooking techniques that they used to cook the foods over 7 consecutive days: 5 weekdays and 2 weekend days. Each participant received oral and written instructions to complete the 7-day record, and all of them received a digital food thermometer. Each of the record sheets was classified into 6 meals (breakfast, morning snack, lunch, afternoon snack, dinner, and other snacks). The volunteers were asked to specify the food or ingredients of a dish, the culinary technique applied, the temperature (°C) of the food or dish at the end of cooking, and the amount eaten (in grams or home measurement). In the case of pre-cooked foods, the brand name was required, and if possible, the product label was also requested.

Criterion validity was assessed by intraclass correlation coefficient (ICC) analysis to establish agreement between the gold standard and the new HCFQ in terms of the use of different cooking techniques. The frequency of use of each cooking technique derived from the HCFQ, or the 7-day food and culinary record was calculated by summing the frequency of use of each cooking technique regardless of food group. The ICC was interpreted as follows: mild agreement (ICC < 0.40), moderate agreement (ICC = 0.40–0.75), and good agreement (ICC > 0.75).

### 2.5. Phase 5: Test–Retest Reliability and Internal Consistency

The test–retest reliability identifies whether the new tool gives the same results when repeated under similar circumstances [39]. A convenience sample of 52 participants was selected, being a similar sample size to other studies [19,22], and data were collected in January 2021. The final version of the HCFQ was formatted online, and a hyperlink to the questionnaire with an access code was emailed to each participant. Volunteers were asked to complete the HCFQ twice, with a 2-week timeframe between the tests, which is considered a short enough time to minimize lifestyle changes and a long enough time to prevent participants from recalling previous responses [40]. Participants included in the reliability phase did not participate in the previous phases of the validation study.

Test–retest was examined using the intraclass correlation coefficient (ICC) for continuous variables and Cohen’s kappa coefficient for categorical variables. ICC values were interpreted as mild agreement (ICC < 0.40), moderate agreement (ICC = 0.40–0.75) and good agreement (ICC > 0.75); and the Cohen’s kappa coefficient as acceptable agreement (κ > 0.40), good agreement (κ > 0.60), and excellent agreement (κ > 0.80) [41]. We grouped some of the items of the new HCFQ. In the case of the domain “use of cooking techniques”, we calculated the frequency of use of each cooking technique as mentioned above. In the domain “use of cooking ingredients”, the items were grouped into 5 subdomains: preparation of different homemade dishes; other ingredients; oils and fats; condiments and spices; and aromatic herbs. Finally, the items of the fifth domain were grouped in a single group of cooking utensils.

To determine the internal consistency of the last version of the questionnaire, Cronbach’s alpha coefficients at time 1 were calculated for each domain of the questionnaire during this phase (acceptable cut-off > 0.70).

## 3. Results

### 3.1. Phase 1: Questionnaire Development

The first version of the questionnaire contained 171 items categorized into five domains: “cooking habits”, “dietary habits”, “use of cooking techniques”, “use of cooking ingredients”, and “use of cooking utensils” (Figure 1). The items of the first domain, “cooking habits”, included different response options ranging from 2 to 8. The “dietary habits” domain comprised nine frequency response options: never or rarely, 1–3 times a month, and 1, 2, 3, 4, 5, and 7 times a week. Items of the domains “use of cooking techniques” and “use of cooking ingredients” included eight frequency response options: never or rarely, 1–3 times a month, 1–2 times a week, 3–4 times a week, 5–6 times a week, once a day, twice a day, and 3 or more times a day. In the domain “use of cooking techniques”, there was the possibility to calculate the frequency of use of each cooking technique by adding up the frequency of use of each technique independently of the food group. Finally, the response options of the items of the domain “use of cooking utensils” were dichotomous (Yes/No). In each domain, participants were asked about the average frequency during the previous year.

### 3.2. Phase 2: Content Validity

Table 1 and Table 2 summarize the S-CVI and I-CVI. The S-CVI demonstrate content validity in terms of both relevance and clarity. The S-CVI/Ave ranged from 0.93 to 1.00 for the domains, exceeding the minimum standard of 0.80. The S-CVI/UA ranged between 0.60 and 0.99 for the domains, exceeding for all domains the minimum standard of 0.80, except the domain “use of cooking utensils” for relevance and the domains “cooking habits”, “dietary habits”, and “use of cooking techniques” for clarity.

The I-CVI values for the criterion of relevance ranged from 0.83 to 1.00, and κ* indices were excellent (κ* 0.75–1.00), except for the domain “use of cooking techniques” in which 1.80% (*n* = 2) of the items had an I-CVI of 0.67 and, according to κ*, were evaluated as fair (κ* = 0.40–0.59). In terms of clarity, the I-CVI values ranged between 0.83 and 1.00 and the κ* indices were excellent (κ* 0.75–1.00), except for the domain “use of cooking techniques”, in which 1.80% (*n* = 2) of the items were fair (I-CVI = 0.67; κ* = 0.40–0.59).

Based on the ratings and comments, syntactic and semantic modifications were made to four items, including the removal of three items with fair relevance and the inclusion of six new items. As a result, the second version of the HCFQ had a total of 174 items.

### 3.3. Phase 3: Face Validity and Inter-Item Reliability

The volunteers of the face validity and inter-item reliability phase were 47.0% women, and age ranged from 23 to 69 years. In addition, 52.9% had a university education (Appendix A).

Face validity was demonstrated through the participants’ responses to the ad hoc questionnaire (Table 3). All subjects agreed that the HCFQ was interesting and useful for learning about their own cooking habits. Although the tool was in general easy to understand, two participants noted that the instructions were not entirely clear. In addition, participants were unsure whether the questions of the domain “use of cooking techniques” referred to their own or their family’s cooking habits. Modifications were made to create more concise, clear and functional instructions for the questionnaire. In relation to the number of questions and response options, one participant pointed out that the frequency options did not take into account seasonal variability in the use of different cooking techniques. In this regard, we included the following sentence in the new version of the questionnaire: “Seasonal variability (summer/winter) should be taken into account. For example, if you use the barbecue 2 times/week only during summer (3 months), the average consumption per year is 1–3 times/month”.

Internal consistency was demonstrated by Cronbach’s alpha parameters (Table 4). Cronbach’s alpha coefficient for the overall questionnaire was 0.94; and for each domain coefficients were also higher than the acceptable minimum of 0.70 (ranged from 0.74–0.91), except for “dietary habits” (Cronbach’s alpha coefficient = 0.56). We decided to keep this domain because these questions reflected the underlying theoretical domain according to the literature and experts’ experience. Thus, no changes were made to the HCFQ.

### 3.4. Phase 4: Criterion Validity

Of the 30 subjects invited to participate, 26 completed the gold standard, of which 18 did so correctly. Therefore, 18 participants were included in the analysis; 55.6% were females, 50% had a university education, and they ranged in age from 43 to 69 years (Appendix A).

The results of the concordance between the gold standard and the HCFQ in relation to the use of different cooking techniques are shown in Table 5. The ICC showed a moderate correlation for grilling (griddle), steaming, frying, raw vegetables, and omelet (ICC between 0.69 and 0.55) and a lower correlation for microwave (ICC = 0.36) and sautéing (ICC = 0.31). For the remaining cooking techniques (baking/roasting, simmering, braising, and battered/breaded and fried) there was a non-statistically significant concordance between the gold standard and the HCFQ.

### 3.5. Phase 5: Test Stability and Internal Consistency

Of the 52 subjects invited to participate, 51 volunteers completed the HCFQ twice, with a mean of 17.6 (SD 4.9) days between tests. The mean age of the participants was 39.8 (12.0), 70.6% were female, and 62.7% were university educated (Appendix A).

The stability coefficients are shown in Table 6 for the categorical variables (κ) and in Figure 2 for the continuous variables (ICC). In the domain “cooking habits”, with respect to the categorical variables, three items showed weak agreement (κ 0.40–0.59), two showed moderate agreement (κ 0.60–0.79), and two showed strong agreement (κ > 0.80) (Table 6).

For the continuous variables, one item showed moderate agreement (ICC = 0.61–0.80), and the rest showed substantial agreement (ICC = 0.81–1.00) (Figure 2). The domain “dietary habits” showed that the ICC ranged from 0.49 (fair agreement) to 0.71 (moderate agreement) (Figure 2). For the domain “use of cooking techniques”, the ICC showed moderate (ICC = 0.45–0.73) and strong agreement (ICC = 0.82–0.88) (Figure 2). The mean scores of the subdomains within the domain “use of cooking ingredients” were 0.79, 0.58, 0.67, 0.53, and 0.87 for preparation of different homemade dishes, use of different ingredients, use of oils and fats, use of seasonings, and use of spices and aromatic herbs, respectively (Figure 2). The domain “use of cooking utensils” showed an ICC of 0.92, suggesting substantial agreement (Figure 2).

Finally, Cronbach’s alpha coefficients were similar to those of phase 3, face validity and internal consistency, although the domains “cooking habits”, “dietary habits”, and “use of cooking utensils” did not reach the acceptable minimum of 0.70 (Table 4). For the overall questionnaire, Cronbach’s alpha value was 0.90. The domains “cooking habits”, “dietary habits”, and “use of cooking utensils” were retained, as they reflected the underlying theoretical domain according to the literature and experts’ experience.

## 4. Discussion

This study aimed to develop and psychometrically test the HCFQ as a mean to measure the frequency of cooking habits at home. Our results indicate that the new HCFQ was considered by experts to have excellent clarity and relevance. The S-CVI-Ave and S-CVI-UA values were above the minimum standard of 0.80, and the I-CVI and κ* for most items were rated as excellent in both clarity and relevance. Participants found the questionnaire to be acceptable and understandable, demonstrating the face validity of the questionnaire. Furthermore, for the overall Cronbach’s alpha was 0.94 in the Phase 3 analysis and 0.90 in Phase 5, values that add further evidence that the responses between questions are well correlated. The ICCs of the items in the domain “use of cooking techniques” between the HCFQ and the gold standard showed a slight and moderate agreement between the methods, which results in a substantial criterion validity of the questionnaire. Finally, the ICCs (continuous variables) and the ƙ values (categorical variables) of the test–retest reliability study confirms the adequate reliability of the new tool.

In the last two decades, new questionnaires have been developed and validated to assess cooking behavior at home. In general, these questionnaires have focused on the study of eating and cooking knowledge, behavior, and skills [18,19,20,21,22,23,24,25,26]. However, only five of these questionnaires have included a limited number of questions related to the frequency of cooking at home [18,19,23,25,26]. The questionnaire developed by Condrasky et al. (2011) [18] includes three questions, and the tools developed by Barton et al. (2011) [19] and Kennedy et al. (2019) [23] include only one question. The Gallup World Poll (GWP) includes six questions about cooking frequency at the individual and household level and the person who cooks at home, while three questions refer to lunch and the remaining three to dinner [25]. Finally, the questionnaire developed and validated by Raber et al. (2021) [26] asks about the use of eight cooking techniques the last time the main meal was prepared at home. Unlike these questionnaires, the HCFQ includes 60 items on the frequency of cooking techniques used in relation to nine food groups during the past year.

Content validity is one of the most important types of validity to ensure congruence between study objectives and data collection instruments [42]. It is a common starting point for questionnaire validation [19,22]. The new HCFQ is based on a robust framework that combines a recent literature review, existing culinary tools [18,19,20,21,22,23,24], and expert consultation. For any new instrument development, it is desirable that the CVI exceed 0.80 [30]. Most S-CVI/Ave and S-CVI/UA values of the HCFQ, among the panel of experts, were above 0.80. Most items showed acceptable I-CVI scores and provided kappa statistical information on the specific degree of agreement beyond chance [29]. In addition, in the face validity study (Phase 3), participants rated the instrument as adequate and relevant for the measurement of frequency of culinary habits. The face validity study adds further confidence that the questionnaire is acceptable and understandable for the general population (subjects of different sex, age, educational level, and cooking habits). Thus, it can be concluded that the new HCFQ has good content and face validity.

Cronbach’s alpha values for the overall HCFQ and each domain in Phases 3 and 5 of the validation process showed that the HCFQ has good internal consistency. Except the domain “dietary habits”, which showed a Cronbach’s alpha coefficient below 0.70 in both analyses. This domain includes eight questions regarding the frequency of eating home-cooked food for lunch or dinner, eating in a restaurant or cafeteria for lunch or dinner, and buying some home-cooked take away food or convenience food (pizza, lasagna, instant soups) for lunch or dinner. Retaining the domain “dietary habits” was important, because these questions are closely related to culinary habits [1]. For example, Monsivais et al. (2014) [43] showed that subjects who spent less than 1 h a day preparing food at home spent more money on food away from home and used fast food restaurants more frequently compared to subjects who spent more time preparing food at home.

In terms of criterion validity, only the domain “use of cooking techniques” was studied due to the lack of a reference method to validate the rest of the domains of the questionnaire [23,44]. Overall, the analysis showed positive and slight to moderate concordance for the use of the different cooking techniques between the HCFQ and the 7-day food and culinary gold standard record. However, for simmering, baking/roasting, microwaving, braising, and battering/breading and frying, no significant concordance was observed. The lack of concordance between methods for some of the analyzed cooking techniques could be attributed to the gold standard used. Although weighted food records represent one of the gold standard tools in dietary assessment, they are not necessarily a good indicator of culinary habits [45]. This is one of the limitations previously stated by other authors in assessing the criterion validity of a questionnaire related to home cooking habits and/or skills [44]. Another reason that could explain the limited or lack or agreement between the gold standard and the new questionnaire for some items could be related to the participants’ knowledge of cooking techniques. We suggest that those subjects who do not usually cook at home could have more difficulties in recognizing the use of certain cooking techniques. Moreover, we acknowledge that the sample size was limited (n = 18). This fact might decrease the statistical power to find concordance between the HCFQ and the gold standard. The lack of agreement between methods could also be explained by the usual tendency to overestimate healthy lifestyles and underestimate unhealthy lifestyles. Therefore, the results of the criterion validity phase should be interpreted with caution. In future studies, it would be interesting to assess the criterion validity against objective measurements such as blood or urine biomarkers. One approach could be to measure compounds, such as AGEs, whose presence in the human body is associated with the type of food consumed and the cooking technique applied [46].

Finally, the reliability tests showed a wide range of ƙ coefficients or ICC coefficients, depending on the nature of the variables. These results are in line with other studies aimed at validating food and/or culinary skill questionnaires, although these studies analyzed the reliability of their tools using Spearman or Pearson correlation tests instead of the ICC test [18,19,22,24]. Overall, our results showed that the HCFQ has the ability to produce a similar score with the same participants under the same conditions over time and is therefore suitable for studying the effectiveness of cooking training interventions.

We acknowledge that the present study has some strengths and limitations that should be mentioned. First, the content validation indices were not reassessed with a second round of expert consultation, as suggested by other authors, due to the limited timeframe [30]. However, a questionnaire consultation was conducted with the target population (face validity) for item revision and the refinement of the tool. Second, although the criterion validity of the new tool was tested, construct validity was not performed. Future studies will be necessary to assess this psychometric property. Third, the use of a convenience sample does not allow for generalizability of the results. Nevertheless, the sample was recruited taking into account gender, age, and cooking habits, which ensured that a broad segment of the population was represented. The slightly higher number of university-educated participants might limit the self-administration of this questionnaire in a less educated population. In addition, different samples of subjects were used at each stage of the validation process (face validity, criterion validity, and reliability). Fourth, the convenience sampling methods used could have resulted in participants having a more favorable attitude towards the construct measured, including the possibility of a social desirability bias, although sincerity was requested at all stages. Similarly, the convenience sample of experts might have influenced agreement on the relevance and clarity of certain items. However, the selection of experts in the field of dietetics, nutrition, gastronomy, and epidemiology can be considered a strength in judging the items of the new tool, as they are the most likely to use similar measurement tools in their professional work. Moreover, the group of experts in the validation process was independent from the group of experts who developed the questionnaire. Fifth, the present study is a pilot study due to the limited sample size in some of the phases of the validation process. Therefore, it will be interesting to assess the validity of the new tool in large sample sizes and also once translated and adapted to other languages and cultures. Sixth, the HCFQ was developed and validated in Spanish. However, since it includes a wide variety of cooking techniques by food group, once translated and validated, it could be useful to evaluate culinary habits in other countries.

The large number of items (174 items) in the questionnaire could be considered a limitation. However, the questionnaire is easy to fill in, as many questions have a dichotomous answer (yes/no) and the same questions on applied cooking techniques are repeated in several sections, classified by food groups. This means that the time taken to complete the questionnaire is not excessively long. We estimated that self-completion of the questionnaire would take around 20 min, although it could be longer depending on the education level of the subject. When the questionnaire was filled by a trained interviewer, the required time was about 15 min. Therefore, we believe that, similar to the food frequency questionnaire (FFQ), the HCFQ can be a valuable tool in nutritional epidemiology to measure cooking techniques used over the long term. From a public health point of view, this may be relevant, as it is necessary to deepen our understanding of what effect not only what we eat but also how we cook has on our health.

In addition, questions related to the cooking techniques used for food preparation would allow a better assessment of the intake of certain hazardous chemicals produced during the cooking process (HAAs, PAHS, nitrosamines, and AGEs). So far, several studies have derived these compounds only from FFQs, although in these questionnaires it is not possible to differentiate between the different cooking techniques used to cook each food group [47,48]. An exception is the FFQ designed by Luevano-Contreras et al. (2013) [49] to measure AGEs in the diet; however, the design of this questionnaire was based on the selection of foods most related to AGE production, and only cooking methods for meats were covered. Further studies are needed to assess whether the combined use of the HCFQ and a validated semi-quantitative FFQ will allow evaluation of the intake of certain compounds whose amounts depend on food processing, such as AGEs, to be assessed more precisely than if only an FFQ or a food record is used. A final necessary step is to investigate the association between these HCFQ-derived chemicals, in combination or not with an FFQ, and different health outcomes.

In conclusion, the new HCFQ is a promising 174-item tool with satisfactory characteristics of validity and reliability. The HCFQ is the first instrument that assesses the frequency of use of different cooking techniques and is suitable for observational studies and interventional studies, including those aimed at improving health status and eating habits through cooking interventions.

## Figures and Tables

**Figure 1 nutrients-14-01136-f001:**
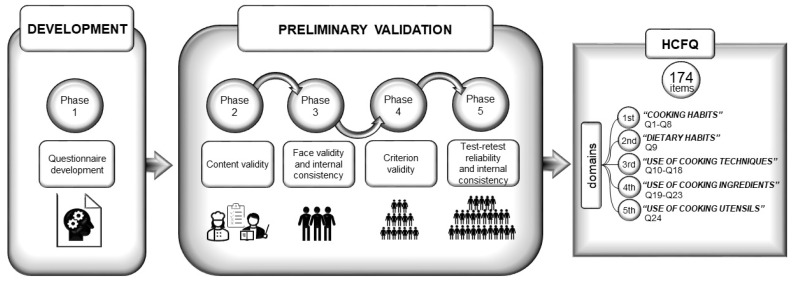
Design of the validation study and graphical description of the Home Cooking Frequency Questionnaire (HCFQ).

**Figure 2 nutrients-14-01136-f002:**
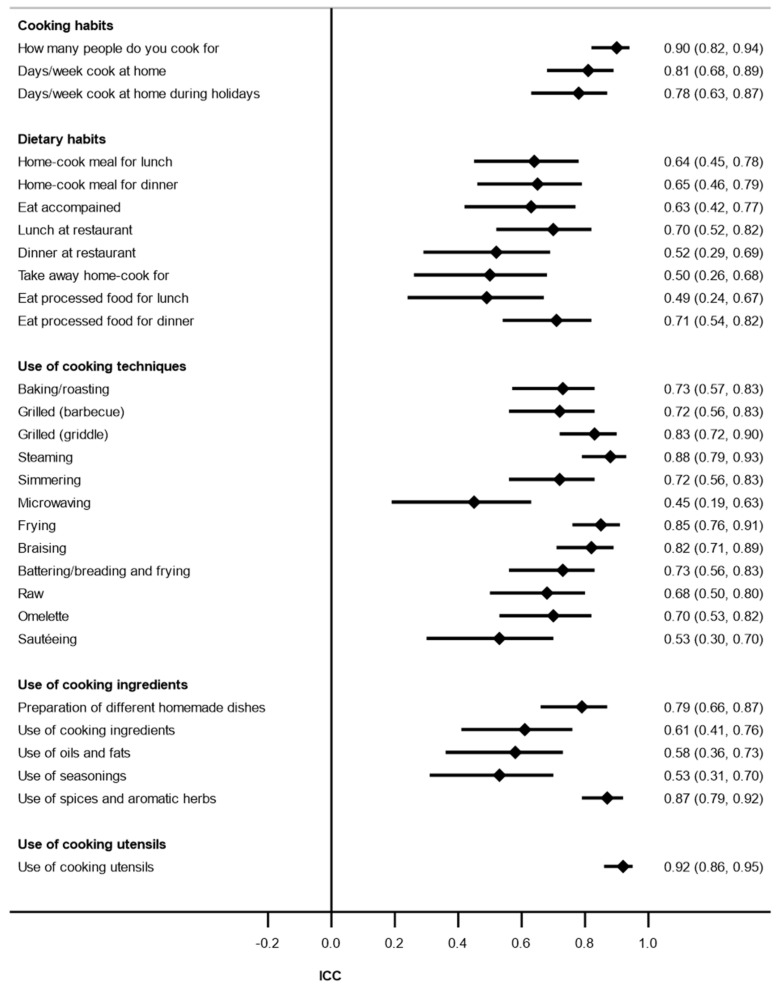
Test–retest reliability for continuous variables (ICC) by domain (test stability).

**Table 1 nutrients-14-01136-t001:** S-CVI-AVE and S-CVI-UA for each domain and the whole questionnaire (*n* = 6) (content validity). S-CVI-Ave indicates the content validity index at scale level with averaging calculation method; S-CVI-UA, content validity index at scale level with the universal agreement method. HCFQ, Home Cooking Frequency Questionnaire.

Domains of the HCFQ	S-CVI-Ave		S-CVI-UA	
	Relevance	Clarity	Relevance	Clarity
Cooking habits	0.98	0.96	0.86	0.79
Dietary habits	0.98	0.93	0.87	0.60
Use of cooking techniques	0.99	0.96	0.93	0.78
Use of cooking ingredients	0.99	1.00	0.97	0.99
Use of cooking utensils	0.96	0.99	0.75	0.93
Whole questionnaire	0.99	0.97	0.93	0.85

**Table 2 nutrients-14-01136-t002:** I-CVI and Kappa* for relevance and clarity of domain and the whole questionnaire (% of items) (*n* = 6) (content validity).

Domains of the HCFQ	I-CVI ^a^ Relevance				κ* for Relevance ^b^				I-CVI ^a^ Clarity				κ* for Clarity ^b^			
	1.00	0.83	0.67	0.5	Excellent	Good	Fair	Poor	1.00	0.83	0.67	0.5	Excellent	Good	Fair	Poor
Cooking habits	85.7	14.3	0	0	100	0	0	0	78.6	21.4	0	0	100	0	0	0
Dietary habits	87.5	12.5	0	0	100	0	0	0	60.0	40.0	0	0	100	0	0	0
Use of cooking techniques	93.3	4.8	1.9	0	98.1	0	1.9	0	78.2	20.0	1.8	0	98.2	0	1.8	0
Use of cooking ingredients	96.7	3.3	0	0	100	0	0	0	98.6	1.4	0	0	100	0	0	0
Use of cooking utensils	75.0	25.0	0	0	100	0	0	0	93.3	6.7	0	0	100	0	0	0
Whole questionnaire	92.8	6.2	1.0	0	99.0	0	1.0	0	84.9	14.2	0.9	0	99.1	0	0.9	0

κ*, modified kappa concordance index. ^a^ I-CVI indicates the content validity index at item level. ^b^ κ* 0.75–1.00 excellent, κ* 0.60–0.74 good, κ* 0.40–0.59 fair, and κ* < 0.40 poor [34].

**Table 3 nutrients-14-01136-t003:** Results of the responses of the ad hoc questionnaire (*n* = 17) (face validity).

	*n* (%)	
	Yes	No
Q1. The questionnaire is interesting	17 (100%)	0 (0%)
Q2. The questionnaire helps you to know your cooking habits	17 (100%)	0 (0%)
Q3. The questionnaire includes all aspects about cooking habits	13 (93%)	1 (7%)
Q4. The questionnaire is too long	3 (18%)	14 (82%)
Q5. The instructions given throughout the questionnaire are well understood	15 (88%)	2 (12%)
Q6. There are questions difficult to understand or answer	1 (7%)	14 (93%)
Q7. The number of response options is adequate and sufficient	15 (88%)	2 (12%)
Q8. Other comments or ideas to improve the questionnaire	2 (12%)	15 (88%)

**Table 4 nutrients-14-01136-t004:** Cronbach alpha coefficient for each domain and the whole questionnaire in Phase 3 and Phase 5 of the validation process (internal consistency).

Domains of the HCFQ	Cronbach Alpha (Phase 3)		Cronbach Alpha (Phase 5)	
	Value	*n*	Value	*n*
Cooking habits	0.76	13–17	0.68	51
Dietary habits	0.56	17	0.63	51
Use of cooking techniques	0.89	14–17	0.76	51
Use of cooking ingredients	0.91	16–17	0.86	51
Use of cooking utensils	0.74	15–17	0.60	51
Whole questionnaire	0.94	13–17	0.90	51

**Table 5 nutrients-14-01136-t005:** ICCs for the use of different cooking techniques between the HCFQ and the 7-day food and culinary record (gold standard) (*n* = 18) (criterion validity).

Cooking Techniques	ICC (95% CI)	*p* Value
Baking/Roasting	0.14 (−0.36; 0.57)	0.285
Grilling (barbecue)	−	−
Grilling (griddle)	0.60 (0.21; 0.82)	0.002
Steaming	0.65 (0.29; 0.85)	0.001
Simmering	−0.06 (−0.38; 0.34)	0.625
Microwave	0.36 (−0.13; 0.70)	0.071
Frying	0.55 (0.07; 0.81)	0.001
Braising	0.09 (−0.41; 0.54)	0.357
Battered/Breaded and fried	0.21 (−0.22; 0.60)	0.176
Raw	0.65 (0.29; 0.85)	0.001
Omelet	0.69 (0.33; 0.87)	0.001
Sautéeing	0.31 (−0.08; 0.65)	0.047

ICC, Intraclass correlation coefficient; 95% CI, 95% confidence interval.

**Table 6 nutrients-14-01136-t006:** Test–retest reliability for categorical variables of the cooking habits domain (ƙ) (*n* = 51) (test stability).

Cooking Habits Domain	ƙ	*p* Value
Plan weekly menus	0.60	<0.001
Weekly grocery shopping	1.00	<0.001
Cook at home	0.92	<0.001
Percentage of dishes cooked at home	0.53	<0.001
Batch cooking	0.44	0.002
Hours/week cooking at home	0.60	<0.001
Hours/week cooking at home during holidays	0.49	<0.001

## Data Availability

Data described in the manuscript, code book, and analytic code will be made available upon request from the corresponding author.

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
