# Peer review of "Development and Validation of a New Home Cooking Frequency Questionnaire: A Pilot Study"

_nutrients, 2022, doi:10.3390/nu14061136_

Round 1
Reviewer 1 Report
Major points:
This manuscript describes the development of a new questionnaire. The purpose of this questionnaire, nor the construct it measures, however are poorly defined. Moreover, the manuscript is riddled with typos and poorly readable. The authors should not just make the changes suggested, but also improve the overall quality of the work. In particular, it should be made clearer how this work can be placed in the wider context and have an English-native person help improve the language.
.
Some overall points to consider:
- For the introduction section, readability could be improved by removing unnecessary details and revising the structure. Generally, start with the overall scope of the topic and clearly identify the problem that you will focus on in this manuscript. Give a clear indication of the theoretical framework you use, and the gap in the literature that your research will fill. Lastly, concretely list the contributions that your paper will make to the state-of-the-art in the field.
- For the discussion section, consider your study's limitations better and focus on both the theoretical and practical (public health) implications of your work.
- Although my comments are not exhaustive, please see below a number of points that may help to improve the study design and manuscript.
Abstract ant title:
L2-3 (title): I don’t think Culinary Habits is a widely used term for the construct you are studying. Suggest to change to Home Cooking Habits.
L20-21: The structure of this sentence is not correct. Suggest rephrasing and remove the comma after “are”.
L21: As Culinary Habits are not a widely used term, please define it first. Moreover, measuring the frequency of a habit (rather than the intensity, difficulty to control, or awareness) seems odd. Please explain.
L23: Porpoise --> Purpose
L23: Why did you use this 5-phase approach? Is this methodology in line with the goal of your study?
L30: Last version? Are there multiple versions?
L31-32: I still miss the relevance of this research and am confused. What do you mean with the frequency of different culinary techniques? Perhaps use an example to clarify.
Introduction:
L46: Reference missing for this statement
L68-71: I am not sure if this rationale merits the development of your questionnaire. It seems there are already well-studied and validated questionnaires out there that are used to measure health outcomes, as you list in your references 10-14
Materials and Methods:
L99-101: Were there also considerations on questionnaire length to this purpose? 174 items seems extremely long
L94-101: Did you define the criteria to which you would measure the to be developed questionnaire prior to developing the questionnaire? If so, please list them clearly. If not, why not?
Discussion & Conclusion:
L289-301: Particularly the Criterion Validity results provide ample room for discussion; please elaborate on them more appropriately.
Author Response
REVIEWER 1
Major points: This manuscript describes the development of a new questionnaire. The purpose of this questionnaire, nor the construct it measures, however are poorly defined. Moreover, the manuscript is riddled with typos and poorly readable. The authors should not just make the changes suggested, but also improve the overall quality of the work. In particular, it should be made clearer how this work can be placed in the wider context and have an English-native person help improve the language.
Thank you for your comment. We have conducted a detailed revision of the manuscript. We have corrected all typos and we have made changes to the quality of the text.
Some overall points to consider:
- For the introduction section, readability could be improved by removing unnecessary details and revising the structure. Generally, start with the overall scope of the topic and clearly identify the problem that you will focus on in this manuscript. Give a clear indication of the theoretical framework you use, and the gap in the literature that your research will fill. Lastly, concretely list the contributions that your paper will make to the state-of-the-art in the field.
Thank you for this comment. We agree that there were unnecessary details and this information has been removed. We have further explained the information provided by previous questionnaires to better justify our questionnaire. Only some of previous questionnaires include questions on cooking frequency and which types of cooking techniques were used, and the number of questions was very limited compared to the new questionnaire we propose. The new paragraph is (lines 72-81): ”A number of questionnaires have been developed and validated to measure food and cooking skills [18–26]. However, these questionnaires are not specifically designed to measure the frequency of exposure to different cooking techniques. In fact, only five out of nine questionnaires asked about cooking frequency and with a low number of questions about this, ranging from 1 to 8, and referring to different time periods [18,19,23,25,26]. Accordingly, it may be relevant to develop a questionnaire that provides a detailed information on how often people cook at home and what cooking techniques are used according to the different food groups (vegetables, meat, fish, eggs, pulses, and cereals).”
- For the discussion section, consider your study's limitations better and focus on both the theoretical and practical (public health) implications of your work.
Following the suggestion of both reviewers, we have expanded the discussion section including more study’s limitations.
- Although my comments are not exhaustive, please see below a number of points that may help to improve the study design and manuscript.
Abstract ant title:
L2-3 (title): I don’t think Culinary Habits is a widely used term for the construct you are studying. Suggest to change to Home Cooking Habits.
Thank you for this suggestion. We have replaced the name “Culinary Habits Frequency Questionnaire” by “Home Cooking Frequency Questionnaire”.
L20-21: The structure of this sentence is not correct. Suggest rephrasing and remove the comma after “are”.
We have rephrased this sentence (lines 20-22): “Home cooking and the type of cooking techniques can affect our health. However, as far as we know, there is no questionnaire that measures in depth the frequency and type of cooking techniques used at home”.
L21: As Culinary Habits are not a widely used term, please define it first. Moreover, measuring the frequency of a habit (rather than the intensity, difficulty to control, or awareness) seems odd. Please explain.
We agree with the reviewer that the term culinary habits may have a different meaning. For this reason, we have opted to use the concept “Home Cooking Frequency” instead of “Culinary Habits”. We think “home cooking frequency questionnaire” better reflects what this new tool is aimed to measure.
L23: Porpoise --> Purpose
We have corrected this typo, thank you for the comment. As we said before, we have revised the whole manuscript to avoid typos and to improve the English style.
L23: Why did you use this 5-phase approach? Is this methodology in line with the goal of your study?
We used a 5-phase approach based on previous questionnaire validations (1-5). This approach permits to evaluate a wide number of psychometric characteristics of the new tool including content validity, face validity, internal consistency, criterion validity, and test stability. At this point it should be noted that as far as we know this is the first time that a questionnaire related to home cooking has been assessed according to the criterion validity.
(1) Rattray J, et al. Essential elements of questionnaire design and development. J Clin Nurs 2007, 16, 234–43.
(2) Carvajal A, et al. How is an instrument for measuring health to be validated ? An Sist Sanit Navar 2011, 34, 63–72.
(3) Streiner DL, et al. Health measurement scales: a practical guide to their development and use; Fifth Edit.; Oxford University Press (OUP): Oxford, 2015.
(4) Schönberg S, et al. Development of the Home Cooking EnviRonment and Equipment Inventory observation form (Home-cookeriTM): An assessment of content validity, face validity, and inter-rater agreement. Nutrients 2020, 12, 1–21.
(5) Kennedy LG, et al. Validity and reliability of a food skills questionnaire. J Nutr Educ Behav 2019, 51, 857–64.
L30: Last version? Are there multiple versions?
Yes, there are different versions that are modified during the validation process. The first version of the questionnaire had 171 items and the last version 174 items (3 items were removed, and 6 included after the validation process). Although the changes in the questionnaire during the validation process are explained along the manuscript, we agree with the reviewer that the use of the concept “last version” in the abstract is confusing. Therefore, in the revised version of the manuscript we have removed this information from the abstract (lines 32-33) “Overall Cronbach’s alpha was 0.90.”
L31-32: I still miss the relevance of this research and am confused. What do you mean with the frequency of different culinary techniques? Perhaps use an example to clarify.
With the frequency of use of different culinary techniques we mean how often foods are cooked at home using different cooking methods. To improve the clarity of this, we have changed the sentence in the revised version of the abstract from “The newly developed HCFQ is an original and tested 174-items tool, being the first to assess the frequency of use of different culinary techniques.” to “This 174-item HCFQ is the first questionnaire to assess how often people cook and which culinary techniques they use at home.” (lines 36-37)
Introduction:
L46: Reference missing for this statement
This information has been removed from the manuscript according to a previous suggestion by the reviewer.
L68-71: I am not sure if this rationale merits the development of your questionnaire. It seems there are already well-studied and validated questionnaires out there that are used to measure health outcomes, as you list in your references 10-14
We appreciate this comment. We have better clarified what our questionnaire adds in comparison with already available validated questionnaires. Most of these questionnaires are focused on cooking knowledge, attitudes or skills and only some of them have a few questions on the frequency of cooking at home. For this reason, we decided to design a new questionnaire following an approach similar to that of the food frequency questionnaire but focused on the culinary techniques used to cook different food groups. We have briefly mentioned this in the introduction and we have further developed this idea at the discussion section.
The new paragraph in the introduction reads as follows (lines 72-81): “A number of questionnaires have been developed and validated to measure food and cooking skills [18–26]. However, these questionnaires are not specifically designed to measure the frequency exposure to different cooking techniques. In fact, only five out of nine questionnaires asked about cooking frequency and with a low number of questions about this, ranging from 1 to 8, and referring to different time periods [18,19,23,25,26]. Accordingly, it may be relevant to develop a questionnaire that provides detailed information on how often people cook at home and what cooking techniques are used to prepare different food groups (vegetables, meat, fish, eggs, pulses, and cereals).” In addition, in the discussion section as (lines 344-365): “In the last two decades, new questionnaires have been developed and validated to assess cooking behavior at home. In general, these questionnaires have focused on the study of eating and cooking knowledge, behavior and skills [18–26]. However, only 5 of these questionnaires include a limited number of questions related to the frequency of cooking at home [18,19,23,25,26]. The questionnaire developed by Condrasky et al. (2011) [18] includes 3 questions, and the tools developed by Barton et al. (2011) [17] and Kennedy et al. (2019) [23] only 1 question. The Gallup World Poll (GWP) includes 6 questions about cooking frequency at the individual and household level, and the person who cooks at home, where 3 questions refer to lunch and the remaining 3 to dinner [25]. Finally, the questionnaire developed and validated by Raber M. et al. (2021) [26] asks about the use of 8 cooking techniques the last time the main meal was prepared at home. Unlike these questionnaires, the HCFQ includes 60 items on the frequency of cooking techniques used in relation to 9 food groups.”
Materials and Methods:
L99-101: Were there also considerations on questionnaire length to this purpose? 174 items seems extremely long
Thank you for this question. Our main objective was to develop a questionnaire that would allow a comprehensive measurement of the cooking techniques used at home for each type of food. We agree that there are many questions but the time it takes to answer them is relatively short. We estimated that self-completion of the questionnaire takes around 20-25 minutes and the completion with a trained interviewer around 15 minutes. This is in part explained by the fact that the response options for some of the items are dichotomous (Yes/No) and for example for the domain “use of culinary techniques”, the list of these techniques is repeated for each food group.
The new paragraph explaining this in the revised manuscript is (lines 473-480): “The large number of items (174 items) in the questionnaire could be considered a limitation. However, the questionnaire is easy to fill in as many questions have a dichotomous answer (yes/no) and the same questions on applied cooking techniques are repeated in several sections, classified by food groups. This means that the time taken to complete the questionnaire is not excessively long. We have estimated that self-completion of the questionnaire can take around 20 minutes, although it could be longer depending on the education level of the subject. When the questionnaire is filled by a trained interviewer, the required time is about 15 minutes.”
L94-101: Did you define the criteria to which you would measure the to be developed questionnaire prior to developing the questionnaire? If so, please list them clearly. If not, why not?
Yes, thank you for this question. We had defined the criteria to develop the questionnaire. These criteria were based on our principal aim of developing a tool able to comprehensively measure the frequency of home cooking and which cooking techniques were used at home for each food group. We have listed these criteria in the revised version of the manuscript (lines 104-108): “The criteria followed to develop the first version of the questionnaire were: to measure the frequency of cooking and consumption of home-cooked food, to identify the main cooking techniques used for each food group and to find out the main cooking habits in relation to aspects such as food shopping and menu planning, the use of different cooking utensils, as well as the spices and types of fats used.”
Discussion & Conclusion:
L289-301: Particularly the Criterion Validity results provide ample room for discussion; please elaborate on them more appropriately.
According to the reviewer’s comment we have expanded the discussion regarding the results provided in the criterion validity analysis (lines 416-424): “Moreover, we acknowledge that the sample size was limited (n=18). This fact might decrease the statistical power to find concordance between the HCFQ and the gold standard. The lack of agreement between methods for the use of some cooking methods, could also be explained by the usual tendency to overestimate healthy lifestyles and underestimate unhealthy lifestyles. Therefore, the results of the criterion validity phase should be interpreted with caution. In future studies, it would be interesting to assess the criterion validity against objective measurements such as blood or urine biomarkers. One approach could be to measure compounds, such as AGEs, whose presence in the human body is associated with the type of food consumed and the culinary technique applied [46].”
Reviewer 2 Report
The purpose of this study is to develop a culinary habit questionnaire and validate it.
In general, the study is well designed, and the manuscript is well written with few exceptions for spelling and grammatic errors.
Here are comments:
Introduction: porpoise should be purpose
Line 36: there should be an ‘a’ before strong, please check the rest of the manuscript for other typos/grammar errors.
Line 38: Use of globally is redundant with Global
Line 51-55 a very long sentence, consider revising
Line 76-78: if the purpose of the newly developed questionnaire to include just frequency of culinary habits, it is not convincing that the whole work was done to test one extra items (the authors states this is lacking from other culinary questionnaires. Also, the authors state in this statement that “evaluate the culinary habits, including the techniques used when cooking at home, to determine their effects in terms of food habits and health”, based on the breakdown of categories, there is no assessment of health characteristics. So, as a reviewer, I don’t think the purpose is convincing.
It would be also good to include the 174-questions as a supplemental file to assess the questions ( the link did not work and was not provided as a separate document).
Although the questionnaire was criterion validated, it is still lacking construct validity to measure is to confirm measurements of theoretical concepts.
Line 404-406 It is not obvious how this questionnaire assesses culinary techniques in relation to health if no questions about health are included.
Also, there is a need to describe the scoring system to better evaluate the usefulness of the questionnaire.
Author Response
REVIEWER 2
Comments and suggestions for authors: The purpose of this study is to develop a culinary habit questionnaire and validate it. In general, the study is well designed, and the manuscript is well written with few exceptions for spelling and grammatic errors. Here are the comments:
Introduction: porpoise should be purpose.
We have corrected this typo, thank you for the comment.
Line 36: there should be an ‘a’ before strong, please check the rest of the manuscript for other typos/grammar errors.
The reviewer is right, the typo at line 36 has been corrected. Moreover, the manuscript has been extensively checked by a native English-speaking colleague.
Line 38: Use of globally is redundant with Global
Thank you for the comment. This sentence has been removed from the manuscript following the suggestion of reviewer 1.
Line 51-55 a very long sentence, consider revising
The reviewer is right, the sentence has been modified (lines 55-58): “Intervention studies aimed to improve cooking skills have shown that participants increase the frequency and variety of the consumption of vegetables and fruits and they experience sizable health improvement in terms of weight control, and normalization of lipid and glucose metabolism biomarkers [2–5].”
Line 76-78: if the purpose of the newly developed questionnaire to include just frequency of culinary habits, it is not convincing that the whole work was done to test one extra items (the authors states this is lacking from other culinary questionnaires). Also, the authors state in this statement that “evaluate the culinary habits, including the techniques used when cooking at home, to determine their effects in terms of food habits and health”, based on the breakdown of categories, there is no assessment of health characteristics. So, as a reviewer, I don’t think the purpose is convincing.
Thank you for the comment. After reviewing each questionnaire related to food skills, cooking skills, cooking frequency and culinary techniques used we realized that none of them was specifically designed to measure the exposure to different culinary techniques which in turn have been associated with health outcomes. In this sense, our aim was to develop and validate a questionnaire that included information about the frequency of used of culinary techniques. In the questionnaire, the domain “use of culinary techniques” is the longest domain of the questionnaire with 60 questions. The total number of questions related to cook frequency is 60 compared to the 8 questions found in the study by Raber M et al. (2022) (1). According to the reviewer’s comment the aim has been changed to “In this pilot study we aimed to develop and preliminary validate a new tool to evaluate the cooking habits, including the techniques used when cooking at home.” (lines 86-88).
(1) Raber M et al. Home cooking quality assessment Tool validation using community science and crowdsourcing approaches. J Nutr Educ Behav 2022.
It would be also good to include the 174-questions as a supplemental file to assess the questions ( the link did not work and was not provided as a separate document).
The questionnaire was included as a supplemental file when we submitted the manuscript to the journal. We are sorry that you were unable to access the supplementary material.
Although the questionnaire was criterion validated, it is still lacking construct validity to measure is to confirm measurements of theoretical concepts.
The reviewer is right, we did not evaluate the construct validity. Therefore, it has been mentioned on the limitations paragraph as follows (lines 442-444): “Second, although the criterion validity of the new tool was tested, construct validity was not performed. Future studies will be necessary to assess this psychometric property.”
Line 404-406 It is not obvious how this questionnaire assesses culinary techniques in relation to health if no questions about health are included.
The reviewer is right, the questionnaire does not include information about health. However, the information obtained through the new questionnaire could be associated with health outcomes collected using other questionnaires. This is possible because there is a relationship between the use of specific culinary techniques and health outcomes. For example in two US cohorts (the Nurses’ Health Study and the Health Professionals Follow-up Study) concluded that frequent fried-food consumption was significantly associated with risk of incident T2D and moderately with incident coronary artery disease (1). In the SUN project (Spanish cohort) demonstrated that fried-food consumption was related to hypertension (2). Moreover, as we explain in the introduction, during cooking process a series of hazardous chemical compounds (advanced glycation end products (AGEs), heterocyclic aromatic amines (HAAs), polycyclic aromatic hydrocarbons (PAHs), and nitrosamines) which can promote inflammation and oxidative stress are formed (3-6). Culinary techniques such as grilling, broiling, roasting, searing, and frying generate more of these compounds compared to boiling, poaching, stewing, and steaming (3,4,7,8). Therefore, it is very interesting to know how often a person use each culinary technique in order to associate this information with health status based on previous scientific literature.
(1) Cahill LE, et al. Fried-food consumption and risk of type 2 diabetes and coronary artery disease: a prospective study in 2 cohorts of US women and men. Am J Clin Nutr. 2014;100:667-75
(2) Sayon-Orea C, et al. Reported fried food consumption and the incidence of hypertension in a Mediterranean cohort: the SUN (Seguimiento Universidad de Navarra) project. Br J Nutr. 2014;112:984-91.
(3) Barzegar F et al. Heterocyclic aromatic amines in cooked food: A review on formation, health risk-toxicology and their analytical techniques. Food Chem. 2019, 280:240–4.
(4) Sampaio GR, et al. Polycyclic aromatic hydrocarbons in foods: Biological effects, legislation, occurrence, analytical methods, and strategies to reduce their formation. Int J Mol Sci. 2021, 22: 6010–40.
(5) Tricker AR et al. Carcinogenic N-nitrosamines in the diet: occurrence, formation, mechanisms and carcinogenic potential. Mutat Res Toxicol 1991, 259:277–89.
(6) Davis KE et al. Advanced glycation end products, inflammation, and chronic metabolic diseases: Links in a chain? Crit Rev Food Sci Nutr. 2016, 56:989–98.
(7) Clarke et al. Dietary advanced glycation end products and risk factors for chronic disease: A systematic review of randomised controlled trials. Nutrients. 2016, 8, 125–51.
(8) Li L, et al. Influence of various cooking methods on the concentrations of volatile N-nitrosamines and biogenic amines in dry-cured sausages. J Food Sci. 2012, 77:560–65.
(9) Uribarri J, et al. Advanced glycation end products in foods and a practical guide to their reduction in the diet. J Am Diet Assoc. 2010, 110: 911–16.
Also, there is a need to describe the scoring system to better evaluate the usefulness of the questionnaire.
We appreciate the reviewer’s suggestion; it would be very interesting to develop a score system to evaluate the results of the questionnaire as a whole. Up to date, in the domain “use of culinary techniques” we are able to derivate how often a person uses each cooking method per day, per week or per month, by adding the specific cooking method in each food group. This information has been included in the revised version of the manuscript (lines 215-217): “In the domain “use of cooking techniques” there is the possibility to calculate the frequency of use of each culinary technique by adding up the frequency of use of each technique independently of the food group.”

Round 2
Reviewer 1 Report
Dear authors,
You have done a great job in improving this manuscript. Thanks for addressing my concerns so thoroughly. I saw one minor point in Figure 2: "People cooks for". Could this be clarified in a more easily understandable term?
Author Response
Dear authors, You have done a great job in improving this manuscript. Thanks for addressing my concerns so thoroughly. I saw one minor point in Figure 2: "People cooks for". Could this be clarified in a more easily understandable term?
Thank you for the comments. The authors appreciate the extensive review of the manuscript.
The reviewer is right, it is not clear the meaning of “People cooks for” in Figure 2. The sentence has been changed to “How many people do you cook for”